# Immunogenicity of BNT162b2 Vaccination against SARS-CoV-2 Omicron Variant and Attitudes toward a COVID-19 Booster Dose among Healthy Thai Adolescents

**DOI:** 10.3390/vaccines10071098

**Published:** 2022-07-08

**Authors:** Pavinee Assavavongwaikit, Napaporn Chantasrisawad, Orawan Himananto, Chayapa Phasomsap, Pintusorn Klawaja, Sapphire Cartledge, Rachaneekorn Nadsasarn, Thidarat Jupimai, Surinda Kawichai, Suvaporn Anugulruengkitt, Thanyawee Puthanakit, on behalf of the Study Team

**Affiliations:** 1Department of Pediatrics, Faculty of Medicine, Chulalongkorn University, Bangkok 10330, Thailand; 6475497630@student.chula.ac.th (P.A.); napaporn.cha@chula.ac.th (N.C.); suvaporn.a@chula.ac.th (S.A.); 2Thai Red Cross Emerging Infectious Diseases Clinical Center, King Chulalongkorn Memorial Hospital, Bangkok 10330, Thailand; 3Center of Excellence for Pediatric Infectious Diseases and Vaccines, Faculty of Medicine, Chulalongkorn University, Bangkok 10330, Thailand; chayapa.p@chula.ac.th (C.P.); pintusorn.k@chula.ac.th (P.K.); rachaneekorn.n@chula.ac.th (R.N.); thidarat.j@chula.ac.th (T.J.); surinda.ka@chula.ac.th (S.K.); 4Monoclonal Antibody Production and Application Research Team, National Center for Genetic Engineering and Biotechnology (BIOTEC), Pathum Thani 12120, Thailand; orawanh@biotec.or.th; 5School of Medicine, University of Birmingham, Birmingham B15 2TT, UK; sxc1091@student.bham.ac.uk

**Keywords:** SARS-CoV-2 vaccine, booster dose, neutralizing antibody titer, anti-SARS-CoV-2 IgG, BNT162b2 vaccine, SARS-CoV-2 Omicron variant, adolescents, vaccine hesitancy

## Abstract

Despite the BNT162b2 vaccination coverage, rapid transmission of Omicron SARS-CoV-2 has occurred, which is suspected to be due to the immune escape of the variant or waning vaccine efficacy of multiple BNT162b2 vaccination doses. Our study aims to compare immunogenicity against Omicron prior to and post a booster dose of BNT162b2 in healthy adolescents, and to evaluate their attitudes toward booster dose vaccination. A cross sectional study was conducted among healthy adolescents aged 12–17 who received two doses of BNT162b2 more than 5 months ago. Participants and their guardians performed self-reported questionnaires regarding reasons for receiving the booster. A 30 ug booster dose of BNT162b2 was offered. Immunogenicity was evaluated by a surrogate virus neutralization test (sVNT) against the Omicron variant, and anti-spike-receptor-binding-domain IgG (anti-S-RBD IgG) taken pre-booster and 14-days post-booster. From March to April 2022, 120 healthy Thai adolescents with a median age of 15 years (IQR 14–16) were enrolled. sVNT against Omicron pre- and post-booster had 11.9 (95%CI 0–23.9) and 94.3 (90.6–97.4) % inhibition. Geometric means (GMs) of anti-S-RBD IgG increased from 837 (728, 953) to 3041 (2893, 3229) BAU/mL. Major reasons to receive the booster vaccination were perceived as vaccine efficacy, reduced risk of spreading infection to family, and safe resumption of social activities. A booster dose of BNT162b2 elicits high immunogenicity against the Omicron variant. Motivation for receiving booster doses is to reduce risk of infection.

## 1. Introduction

As of 29 May 2022, there have been more than 525 million cases of COVID-19 worldwide with more than 6.2 million deaths [1]. In Thailand, over 4.4 million cases of COVID-19 have been reported including 744,000 cases in children or adolescents [2]. This pandemic has impacts on children, not only physical, but also on social, educational, and emotional well-being [3]. The BNT162b2 COVID-19 vaccine, which contains nucleoside modified messenger RNA encoding the severe acute respiratory syndrome coronavirus 2 (SARS-CoV-2) spike glycoprotein, is recommended for healthy adolescents globally. Data from a pivotal trial of the BNT162b2 COVID-19 vaccine in adolescents showed that adolescents have a higher immune response compared with young adults; the geometric means (GMs) ratio of SARS-CoV2 geometric neutralizing mean titers was 1.76-fold higher in adolescents [4]. Effectiveness of two doses of the BNT162b2 vaccine against COVID-19 hospitalization was 93% during the Delta predominant period [5].

Since the emergence of SARS-CoV-2 Omicron variants in December 2021, multiple reports have described reduced effectiveness of BNT162b2 against SARS-CoV-2 infection to 38%, though efficacy against hospitalization and severe disease remain high [6,7,8,9]. Reduction in vaccine effectiveness against SARS-CoV-2 infections is explained primarily due to the variant escaping vaccine protection and the waning of immunity over time [10,11]. Studies evaluating vaccine effectiveness (VE) against symptomatic disease found that VE was significantly reduced in the Omicron era compared to the Delta predominant period [12,13,14,15]. Therefore, several countries such as the USA recommend a booster dose for adolescents ≥12 years old and adults, with an interval at least 5 months after the second primary series dose [6,10,16].

A successful vaccine needs high effectiveness, safety, and high acceptance rates. To increase vaccine acceptance, one must understand facilitators and barriers behind acceptance of a vaccine. Although there is a plethora of information regarding vaccine uptake and acceptance in adults, little is known about adolescent and parental hesitancy and underlying intentions for COVID-19 vaccines [17,18]. Of the studies that do exist, most examine vaccine acceptance rates and hesitancy, and were carried out before vaccines were authorized for use in adolescents [19,20,21,22,23,24]. Additionally, these studies focus on the primary series of vaccination, not booster doses. Since guardians are primary decision makers in adolescents’ medical care, it is important to study guardians’ reasons for wanting their child to receive a booster dose. Additionally, in some European countries, including in the United Kingdom (UK), adolescents under 16 years are able to make decisions regarding their medical treatment, including vaccinations, if they are deemed competent. It is important to understand underlying intentions for receiving the booster dose in both adolescents and their guardians to shape future communication, and to build trust in vaccines and thus increase vaccine acceptance.

This study aims to compare immunogenicity responses pre and post a booster dose of BNT162b2, and attitudes of adolescents and their guardians toward booster dose vaccination administration in adolescents.

## 2. Materials and Methods

### 2.1. Study Design and Participants

This study is a cross sectional study among 120 healthy adolescents, aged 12–17 years old, who received two doses of BNT162b2 more than 5 months before study commencement, which was conducted at the Faculty of Medicine, Chulalongkorn University, Bangkok, Thailand. Adolescents who had history of receiving blood products within 3 months, any vaccines (within 2 weeks for inactivated vaccines or 4 weeks for lived vaccines), or with previous SARS-CoV-2 infection were not eligible. Signed written informed assent and consent were obtained from eligible adolescents and their guardians who agreed to participate prior to study enrollment.

This study was registered in the Thai Clinical Trials Registry (thaiclinicaltrials.org, TCTR20220301002) and approved by the Institutional Review Board of the Faculty of Medicine, Chulalongkorn University (IRB No. 01/65).

### 2.2. Study Procedures

The adolescents who were eligible and consented to participate were enrolled. A 30 ug booster dose of BNT162b2 was offered to all adolescents. The booster vaccination was provided if both the adolescent and their guardian accepted. The first 60 participants by enrollment order had blood collection performed prior to getting the booster dose (pre-booster). Participants in this group were also offered blood collection at 2–4 weeks after the booster dose (post-booster) of which 31 participants agreed to. In the other 60 participants, blood collection was performed 2–4 weeks after the booster dose (post-booster).

All samples were tested for anti-spike-receptor-binding-domain IgG (anti-S-RBD IgG) and a surrogate virus neutralization test (sVNT) against B.1.617.2 (Delta variant) and B.1.1.529 (Omicron variant) was performed. Paired samples of pre- and post-booster were also tested with a pseudo virus neutralization test (pVNT) against B.1.1.529 (Omicron variant).

All participants and their guardians performed self-reported questionnaires regarding their attitudes toward COVID-19 infection and vaccination, and chose their top 3 reasons out of 15 reasons affecting the acceptability of the COVID-19 booster-dose vaccine before they got the booster vaccination. In the pre-booster group, the questionnaires were performed before blood collection. In the post-booster group, the questionnaires were performed before getting the booster vaccination and then an appointment for blood collection subsequently followed.

### 2.3. Immunogenicity Measurement

#### 2.3.1. Quantitative Anti-Spike-Receptor-Binding-Domain IgG (anti-S-RBD IgG) ELISA

The ELISA protocol was adapted from Amanat et al. [25] and performed as described previously [26]. The ELISA plates were coated with purified recombinant Myc-His-tagged S-RBD, residues 319–541 from SARS-CoV-2 (Wuhan-Hu-1). Anti-S-RBD IgG level was reported in binding-antibody units (BAU/mL) following conversion of OD450 values with the standard curve using known units of WHO international standard (NIBSC 20/136).

#### 2.3.2. Surrogate Virus Neutralization Test (sVNT)

A surrogate virus neutralization test was set up as previously described in Tan et al. [27] and our previous work [26]. Recombinant S-RBD (residues 319–541) from Delta (B.1.617.2) or Omicron (B.1.1.529; BA.1) strains and the ectodomain of human ACE2 were purified from Human Embryonic Kidney (HEK) 293T cells. Serum samples (at 1:10 dilution)—S-RBD mixture were incubated in 96-well plates coated with 0.1 µg/well recombinant human ACE2 ectodomain. Then, ELISA was performed. The negative sample was pre-2019 human serum. The % inhibition was calculated as follows:$$\%inhibition=100\times \left[1-\frac{sampleOD450}{negativeOD450}\right]$$

#### 2.3.3. Pseudo Virus Neutralization Test (pVNT)

The pseudo virus neutralization test (pVNT) against the Omicron variant was performed as described previously [28]. Twofold serial dilutions of sera (starting 1:40) were incubated with pseudoviruses displaying the Omicron (B.1.1.529; BA.2) spike in a 1:1 vol/vol ratio in a 96-well culture plate for 1 h at 37 °C. Subsequently, suspensions of HEK293T-ACE-2 cells (2 × 104 cell/mL) were mixed with the serum–pseudovirus mixture and seeded into each well. The neutralizing antibodies were determined based on luciferase activity. Values were normalized against signals from no-serum controls. The pVNT50 values were calculated by determining the half-maximal inhibitory dilution.

### 2.4. Questionnaire Development

We modified the Vaccine Hesitancy Scale, an instrument developed by WHO’s Strategic Advisory Group of Experts on Immunization in 2015 [29] which has been used in numerous countries, to assess hesitancy among guardians for childhood or adolescent vaccines. We also adapted questions from previous vaccine surveys [30] and added new questions which could be potential influencing factors specifically for COVID-19 vaccines.

Overall, questionnaires for adolescents and guardians mainly asked about COVID-19 disease and vaccination. Questionnaires included demographic data along with questions using Likert scales (1–5 scores) and multiple choices. Participants gave answers relating to perceptions of COVID-19 disease and infections [23] on a Likert scale from 1 to 5 with 1–2 being ‘disagree’, 3 being ‘uncertain’, and 4–5 being ‘agree’. Multiple choice answers were used to assess reasoning for receiving a booster dose vaccination; 15 answers were displayed, and participants chose their top 3 answers. We adapted the 15 choices from a previous study [31,32] to make it more applicable to the Thai population.

### 2.5. Statistical Analysis

Descriptive statistics were utilized for data analysis in this study. Categorical variables were presented with absolute numbers and percentages, and continuous variables with medians and interquartile range (IQR) or mean with standard deviation (SD) or 95% confidence interval (CI). A Z-test for proportions, *t*-test, and median test were used to determine statistically significant differences, where appropriate, with an alpha-value of 0.05 as statistically significant cut off point. The anti-S-RBD IgG was transformed to a natural log scale to perform statistical analysis and converted back to the original scale to report. Stata/SE 13.0 was used for data analyses.

## 3. Results

### 3.1. Study Populations

During March to April 2022, 120 adolescents were enrolled in our study and 116 adolescents agreed to receive the booster vaccination (96.7%). The median (IQR) age of adolescents was 15 (14,16) years, 67 (55.8%) were female, and 80 (66.7%) studying at the high school level. Of the guardians, 108 (90%) were aged 40 years or older and 116 (97%) were the adolescents’ biological mother or father. Most of the guardians had a bachelor’s degree or higher (93%) and 88 (73%) reported household income was more than USD 1500 (≥50,000 Baht) per month. Nineteen (16%) and twenty-eight (23%) of these households reported the experience against infection and getting isolation due to COVID-19 among their family members, respectively. Baseline characteristics of study participants are shown in Table 1.

### 3.2. Comparison of Immunogenicity against SARS-CoV-2 Delta and Omicron Strains between Pre- and Post-Third Dose of BNT162b2

The median (IQR) time between second dose of BNT162b2 and the 60 blood samples drawn for laboratory tests for pre-booster was 5.4 (5.1, 6.0) months. The geometric mean (GM) of anti-S-RBD IgG (95% CI) was 837 (728–953), the median (IQR) % inhibition for sVNT against the Omicron variant was 11.9 (0–23.9), and the median (IQR) % inhibition for sVNT against the Delta variant was 82.9 (64.1–95.6). The median (IQR) time between third dose booster vaccination and blood drawn to measure antibody response post-booster was 14 (14–15) days. The GM of anti-S-RBD IgG (95% CI) was 3041 (2893–3229), the median (IQR) % inhibition for sVNT against the Omicron variant was 94.4 (90.6–97.4), and the median (IQR) % inhibition for sVNT against the Delta variant was 100.0 (99.9–100.0). All of the immunogenicity components measured—the GM of anti-S-RBD IgG, % inhibition for sVNT against the Omicron variant, and % inhibition for sVNT against the Delta variant—prior to and 14 days after the booster dose exhibited very high statistical significance where all *p*-values were <0.001 (Table 2).

To eliminate inter-personal variations, we compared the lab results among 31 participants who had both pre- and post-booster vaccination paired serum; the immunogenicity response is shown in Table 2. The results confirmed the result from the unpaired comparison where all components were significantly different and the post-booster values were higher than the pre-booster. The median (IQR) ratios of post-booster and pre-booster of anti-S-RBD IgG among these 31 paired samples was 3.8-fold (2.9, 4.7), (*p* < 0.001).

For the 31 samples in the paired group, pVNT against the Omicron variant was performed. The results showed that, post-booster vaccination, neutralizing antibodies against the Omicron variant could be detected at a very high level with a pVNT50 value of 912 (622–1507).

### 3.3. Attitudes toward COVID-19 Infection and Factors Involved in Getting COVID-19 Booster Vaccination of Adolescents and Their Parents

The attitudes toward COVID-19 infection and vaccination of adolescents and their parents are summarized in Table 3. Perception of COVID-19 as a serious disease among the guardians was 88.3% compared to 73.3% among adolescents (*p* < 0.01). Guardians more actively searched for COVID-19 vaccine information (85.8%) and thought that the information about COVID-19 vaccines they received was reliable (90.0%), when compared to adolescents at 53.3% (*p* < 0.01) and 80.0% (*p* < 0.05), respectively. Overall, 98.3% of guardians intended for their children to get booster vaccinations against COVID-19, while 89.2% of adolescents intended to get vaccination for themselves (*p* < 0.01). Similar proportions of guardians and adolescents thought that COVID-19 booster vaccines are necessary for children and adolescents (94.2% vs. 98.3%, *p* > 0.05). There was no significant difference between guardians’ and the adolescents’ opinion about COVID-19 vaccines safety and efficacy in preventing infection.

Table 4 outlines the reasons adolescents gave for receiving the booster dose, and reasons guardians gave for wanting their child to have a booster dose. Participants chose their top 3 reasons from the list of 15 reasons in Table 4. The top reason for booster vaccination chosen by guardians (60.8%) and adolescents (53.3%) was the perception that the booster vaccine has a high efficacy. Other highly chosen reasons were the perceived safety of the COVID-19 vaccine with 58.0% of guardians and 45.8% of adolescents selecting this option, and to prevent spread to family and friends with 54.2% of guardians and 48.3% of adolescents selecting this option. When comparing adolescents’ and guardians’ reasons for receiving the booster dose, we found that there was a statistically significant difference for the reason to reduce community spread between adolescents and guardians (20.8% of adolescents, 10.8% of guardians; *p* = 0.03). We decided to group two options together: ‘family or friends’ recommendation’ and ‘people in community obtaining COVID-19 vaccination for adolescents’ and combined them into ‘influences from others’ as we found a highly significant difference for influences from others as a higher choice in adolescents, compared to guardians (14.2% of adolescents, 1.7% of guardians; *p* < 0.01).

## 4. Discussion

Our study found that a 30 µg booster dose of BNT162b2 administered to adolescents aged 12–17 years elicited a high immunogenicity response against the Omicron variant which translated to a median surrogate virus neutralization test (sVNT) of 94.4 (IQR 90.6, 97.4) % inhibition in post-booster samples compared to 11.9 (IQR 0, 23.9) % inhibition in pre-booster samples. Evaluation of reasoning behind booster vaccine acceptance highlighted the most important reason for both adolescents receiving the booster and guardians wanting their child to receive the booster were the perceived high efficacy of the booster vaccination to prevent infection (53.3% in adolescents; 60.8% in guardians), to prevent spread of COVID-19 to family and friends (48.3% in adolscents;54.2% in guardians), and the perceived safety of the booster vaccine (45.8% in adolescents; 58.0% in guardians).

Studies in adults demonstrated a higher immunogenicity against Omicron following a booster dose of BNT162b2 when compared to a two-dose regimen [16,33,34,35,36]. Nemet et al. [35] found, in adults following receipt of a third dose of BNT162b2, neutralization against Omicron increased from geometric mean (GM) titer 1.11 to 107.6, which aligns with our study findings in adolescents. Previous studies evaluating BNT162b2 booster doses against Omicron in adolescents have primarily studied vaccine effectiveness (VE) and show significant increase in VE post-booster [6,37,38]. Our study did not evaluate VE; however, these studies can be considered together with our study findings to inform policy makers when deciding whether it might be applicable to administer booster doses in adolescents, and their guardians when deciding whether they will receive a booster dose for their children.

Our study found that Omicron elicits vaccine escape from neutralizing antibodies induced by the primary vaccination series. Among these pre-booster samples, the median sVNT against Omicron was 11.9% inhibition, while against Delta strains it remained as high as 82.9% inhibition. A decreased immunogenicity of BNT162b2 against Omicron in adolescents has been observed in similar studies [10,39]. Chen et al. [39] found following a primary series of BNT162b2 vaccination among those aged 12–18, geometric mean microneutralization antibody titer against Omicron was 7.2. This finding also aligns with the conclusion of studies in adults [16,33,34,35,36,40]. Therefore, this shows the decreased susceptibility of Omicron in adolescents following two doses of BNT162b2.

Globally, there is disparity in approaches taken by countries regarding vaccination of children and adolescents. The Centers for Disease Control and Prevention (CDC, Atlanta, GA, USA) [41] recommends everyone over the age of 5 have a booster dose, whereas the Joint Committee on Vaccination and Immunization (JCVI, UK) [42] recommends adolescents aged 16–17 years old have a booster dose, or adolescents aged 11–15 if they fulfill certain criteria, such as being immunocompromised. The European Medicines Authority (EMA, Europe) [43] recommends boosters for those aged above 12, with different countries in Europe devising their own strategies. Therefore, different countries have unique approaches which demonstrates a ‘not one size fits all approach’. Our study can be used as evidence for increased immunogenicity against Omicron following a BNT62b2 booster dose and considered by policy makers to aid decisions as to whether to provide booster vaccinations for adolescents.

Our study found that the major underlying intentions adolescents had to receive the booster vaccine, and guardians of adolescents had for their child to receive a booster dose, were the perceived high efficacy of the vaccination to prevent infection, prevention of spread of COVID-19 to family and friends, and the perceived safety of the booster vaccine. A recent systematic review and meta-analysis investigated parents’ and guardians’ underlying reasons to vaccinate their child with a COVID-19 vaccine [44]. The primary reason was to protect their children and those around them. A study in Hong Kong [19] evaluating adolescent intention to receive the COVID-19 vaccination found the major reasons were that they were worried about infection, wanted to protect their family, and wanted to return to normality before COVID-19. Thus, we see a theme emerging that a major reason adolescents want, and their guardians want them, to receive the booster is to protect themselves and those around them. This suggests adolescents have a high level of knowledge regarding the COVID-19 vaccine as they understand the vaccine reduces infection spread thus protecting those more vulnerable around them. This can allow policy makers to tailor public health strategies around this level of high competency to adolescents when aiming to increase vaccine acceptance.

When evaluating motivational reasons for booster vaccine receipt between adolescents and guardians wanting their child to be vaccinated, we found one choice to be highly statistically significantly different. Here, 14.2% of adolescents compared to 1.7% of guardians (*p* < 0.001) stated their reason for a booster dose was the influence from those around them such as friends, family, and the greater community. Several theories exist that hypothesize adolescent decision making is different to that in adults due to their decision-making process being heavily modulated by social influence [45,46]. Therefore, adequate support should be offered to enhance knowledge levels in schools and in the community to provide an adequate environment which influences adolescents to make decisions competently.

The strengths of this study include that we focused on adolescents who are a key population at this time following recent announcements internationally authorizing booster doses in this population. Additionally, we used laboratory markers—sVNT and pVNT—and determined the immunogenicity of a BNT162b2 booster against Omicron, the dominant strain at the time we carried out this study. Additionally, we evaluated both adolescent and guardian underlying intentions to get the booster vaccination.

Limitations include that we used an in-house surrogated viral neutralizing antibody rather than the conventional or pseudo virus neutralization test as we tested the pVNT only in post-booster samples of the paired group. However, this test has been trialed and revealed good correlation with conventional and pseudo virus-based methods. Our acceptance rates of the vaccine might have an underlying bias as almost all of study participants knew they would receive a booster dose in this study and therefore those not wanting a booster dose at the time did not sign up and were not included in our study. This also translates to their intentions for getting the vaccine as they might already think the vaccine is safe and effective to prevent infection before signing up to the study. Furthermore, socio-demographic data show that participants came from higher income households with parents who attained a high degree of education. Here, 67.5% of participants had a household income of > USD 1500 a month, and 73.3% of parents were educated to bachelor’s degree level. Therefore, this might have influenced vaccine acceptability and underlying intentions for vaccination and might question the applicability of the findings to a general population.

## 5. Conclusions

Adolescents aged 12–17 years elicited a high immunogenicity response against the Omicron variant following a booster dose of BNT162b2. The uppermost underlying intentions for vaccination with a booster were: perception of high efficacy, perception of high safety, and to reduce spread to others. Our study can be used as evidence to contribute to a decision on whether to vaccinate adolescents with a booster dose on a wide population-based scale for policy makers, and on an individual basis for adolescents and their guardians. Additionally, our findings regarding underlying intentions to vaccinate can aid policy makers when devising public health strategies to increase uptake of the booster vaccine in adolescents.

## Figures and Tables

**Table 1 vaccines-10-01098-t001:** Baseline characteristics of study participants.

	N (%)
Median age of adolescents (IQR)	15 (14, 16)
Adolescents age group	
12–15 years old	73 (60.8)
16–17 years old	47 (39.2)
Male	53 (44.2)
Median age of guardians (IQR)	47 (44, 50)
Relation to the adolescents	
Mother	99 (82.5)
Father	17 (14.2)
Others	4 (3.3)
Education	
Master’s degree or above	31 (25.8)
Bachelor’s degree	81 (67.5)
Non-university certificate or diploma (Vocational school)	3 (2.5)
≤High school	5 (4.2)
Household income (per month) ^a^	
USD 1500 or more (high)	88 (73.3)
Between USD 750–1500 (middle)	30 (25.0)
USD 750 or less (low)	2 (1.7)
Family structure	
Median number of household members (IQR)	4 (4, 5)
Family members vulnerable to severe COVID-19	18 (15.0)
Burden of COVID-19	
Experience of COVID-19 infection in the family	19 (15.8)
Experience of high-risk contact with COVID19	28 (23.3)

^a^ Household income was converted from Thai Baht (approximately 1 Baht to 0.03 USD) to USD and categorized into high, middle, and low income according to Thai population data.

**Table 2 vaccines-10-01098-t002:** Immunogenicity outcomes in healthy adolescents prior to and post receiving a BNT162b2 mRNA vaccine booster dose.

	Pre-Booster (*n* = 60)	Post-Booster (*n* = 91)	*p*-Value
Anti-S-RBD IgG: Geometric means (95% CI)	837 (728, 953)	3041 (2893, 3229)	<0.001 ^a^
sVNT Delta: Median (IQR)	82.9 (64.1, 95.6)	100.0 (99.9, 100.1)	<0.001 ^b^
sVNT Omicron: Median (IQR)	11.9 (0, 23.9)	94.4 (90.6, 97.4)	<0.001 ^b^
	Pre-Booster (*n* = 31)	Post-Booster (*n* = 31)	*p*-Value
Anti-S-RBD IgG: Geometric means (95% CI)	854 (713, 1022)	3134 (2893, 3395)	<0.001 ^c^
sVNT Delta: Median (IQR)	84.2 (63.4, 95.2)	99.9 (99.8, 100.0)	<0.001 ^b^
sVNT Omicron: Median (IQR)	16.7 (0, 20.9)	94.4 (91.2, 97.6)	<0.001 ^b^
pVNT Omicron: Median (IQR)	-	912 (622, 1507)	NA

^a^ *p*-value for *t*-test for means on log scale of Anti-S-RBD IgG; ^b^ *p*-value for median test; ^c^ *p*-value for paired *t*-test for means on log scale of Anti-S-RBD IgG; anti-S-RBD IgG = anti-spike-receptor-binding-domain IgG; sVNT = surrogate virus neutralization test; pVNT = pseudo virus neutralization test; NA = not applicable.

**Table 3 vaccines-10-01098-t003:** Guardians’ and adolescents’ attitudes toward COVID-19 disease and COVID-19 vaccines.

Descriptions	Guardians, N (%, 95% CIs)	Adolescents, N (%, 95% CIs)	*p*-Value ^a^
Burden of COVID-19 disease			
I think COVID-19 is a serious disease	106 (88.3%, 82.6–94.1)	88(73.3%, 65.4–81.2)	<0.01
My family or I could get COVID-19	92 (76.7%, 69.1–84.2)	92 (76.7%, 69.1–84.2)	1.00
There are members in my family who can have a severe disease course if they get COVID-19	54 (45.0%, 36.1–53.9)	51 (42.5%, 33.7–51.3)	0.70
Attitudes toward COVID-19 vaccines			
I search for information about COVID-19 vaccines actively	103 (85.8%, 79.6–92.1)	64 (53.3%, 44.4–62.3)	<0.01
I think that information about COVID-19 vaccines I received are reliable	108 (90.0%, 84.6–95.4)	96 (80.0%, 72.8–87.2)	0.03
I think that COVID-19 vaccines are preventive	75 (62.5%, 53.8–71.2)	87 (72.5%, 64.5–89.2)	0.10
I think that COVID-19 vaccines are safe	91 (75.8%, 68.2–83.5)	96 (80%, 72.8–87.2)	0.44
I think that COVID-19 booster vaccines are necessary for children and adolescents	113 (94.2%, 90.0–98.4)	118 (98.3%, 96.0–100)	0.09
If a booster vaccine against COVID-19 was safe and available as in adults, I would let my children get vaccination	118 (98.3%, 96.0–100)	107 (89.2%, 83.6–94.7)	<0.01

^a^ *p*-value for z-test for proportions between guardians and adolescents.

**Table 4 vaccines-10-01098-t004:** Top three reasons underlying intention to receive the BNT162b2 booster dose in guardians and adolescents.

Descriptions	Guardians, N (%)	Adolescents, N (%)	*p*-Value ^a^
Efficacy and Safety of COVID-19 Vaccine			
Perceived high efficacy of COVID-19 vaccine booster dose to prevent infection	73 (60.8)	64 (53.3)	0.24
Perceived safety of COVID-19 vaccine	69 (58.0)	55 (45.8)	0.07
Prevention of disease spreading			
Prevent spread to family and friends	65 (54.2)	58 (48.3)	0.37
Vaccination can stop the pandemic	25 (20.8)	23 (19.2)	0.75
Reduce community spread	13 (10.8)	25 (20.8)	0.03
To resume normal daily life			
Resume/Increase social activities	51 (42.5)	46 (38.3)	0.51
Able to go back to school	24 (20.0)	18 (15.0)	0.31
Able to travel	5 (4.2)	13 (10.8)	0.05
School requirement	5 (4.2)	5 (4.2)	1.00
Influencing from family and the community			
Family or friends’ recommendation	0	13 (10.8)	-
People in community obtaining COVID-19 vaccination for adolescents	2 (1.7)	6 (5)	1.00
Burden of COVID-19 disease			
Large increase in local COVID-19 cases	12 (10.0)	16 (13.3)	0.42
Knowing someone who became seriously ill or died from COVID-19	2 (1.7)	3 (2.5)	0.65
General recommendation			
Health care professional recommendation	9 (7.5)	7 (5.8)	0.61
Government recommendation	2 (1.7)	1 (0.8)	0.56

^a^*p*-value for z-test for proportions between guardians and adolescents.

## Data Availability

The data supporting this study’s findings are available from the corresponding author upon reasonable request.

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
