# Peer review of "Immunogenicity of BNT162b2 Vaccination against SARS-CoV-2 Omicron Variant and Attitudes toward a COVID-19 Booster Dose among Healthy Thai Adolescents"

_vaccines, 2022, doi:10.3390/vaccines10071098_

Round 1

Reviewer 1 Report

This manuscript describes the results of a cross sectional study in Thailand in which COVID-19 vaccinated adolescents were 1) polled about their attitudes about receiving BNT162b2 vaccine boosters and 2) had their blood assayed for immune reactivity to the Omicron SARS-CoV-2 variant; some subjects had blood drawn before and after a booster shot and some only after a booster shot. The key thing the authors are trying to do is report on Omicron immune reactivity in adolescents after a BNT162b2 booster.

Abstract:

The first line of the abstract is misleading. A more appropriate way of communicating this information would be to say something akin to “Spread of the omicron SARS-CoV-2 variant despite extensive vaccination suggests some degree of immune escape or waning effectiveness of multiple BNT162b2 vaccinations.” This line is particularly suspect given the results of the data in the paper showing statistically significant immune responses to Omicron after a BNT162b2 booster shot in Thai adolescents.

Introduction:

The last two sentences of the second paragraph (line 60-62) can be removed. This research does not directly address either rapid transmission of COVID-19 nor public health interventions.

The use of the word ‘uptake’ in the first two sentences of the third paragraph is not clear. It appears the authors intend to introduce the idea of ‘public acceptance’ rather than some sort of ‘recognition by the immune system’. This may be a result of translation to English?

Materials and Methods:

It is not clear why the participants were divided into two groups (Study Procedures line one)? Were the groups randomly assigned or matched on some group characteristic? One group appears to have had two blood extractions (before and after boost) and one group only one blood collection (after boost). This point is particularly relevant for interpretation of the results. For example, Table 1 provides baseline characteristics of the study participants but does not subdivide the participants into the two blood collection groups. This apparent combination of participants subgroups appears throughout the manuscript where the survey results are reported.

In regard to the self-reported questionnaire, it is not clear WHEN RELATIVE TO RECEIVING THE BOOSTER the self-reported survey was administered. This is significant for understanding the results. Perhaps the adolescents (or their guardians?) thought they would only receive the booster if they answered the questions in a certain way? Perhaps, the respondents answered in a certain way after receiving the booster when they realized the booster did not negatively influence their health, etc. Relatedly, there is no way from the information provided to confirm who answered the questions on the survey. Did the adolescents answer the questions? Did the guardians answer the questions? Did the guardians prompt the adolescents to answer in certain ways? These methodological concerns make the results of the survey difficult to interpret.

Results:

See above for comment on results presented in Table 1.

The immunological results presented in Table 2 are confusing in regard to the division of the subjects into the two subgroups. The top three rows appear to combine all participants and the bottom three rows appears to show the data from the subjects who received both pre- and post-booster blood draws.

The presentation of the results in Tables 3 and 4 are complicated as a result of the division of the subjects into two ‘treatment’ categories. The reported results appear to COMBINE the ‘after’ assays from the two groups of participants (based on the sample sizes shown in the tables). As a result, the study design appears to be some sort of combination of a between- and within-subjects design. This makes the survey and the immunological data presented difficult to interpret.

The results provided do not address the possibility that the results of the survey might be influenced (correlated) with any one or more of the parameters measured in the immunological assays.

No analyses are provided to address the possibilities that male and female adolescents (or other parameter measured in the survey) exhibited similar responses on the survey and/or similar immunological responses on the assays performed. In other words, is there something measured by the survey that ‘predicted’ immune reactivity in any way?

Discussion:

Paragraph 6 of the Discussion would remove the phrase ‘due to their developing brain’. This statement could be objectionable to psychologists and neuroscientists, alike. The paper provides no connection between the responses on a single, CROSS SECTIONAL survey and brain development.

In regard to the data presented potentially informing public policy, it is not important what adolescents’ reasoning might be for getting COVID-19 boosters so long as they get the booster shots. The authors seem to have some bias that underlying intentions to vaccinate/boost might somehow influence the efficacy of the vaccination/boost.

Reviewer 2 Report

This is well written and interesting manuscript and provides detailed information on both the biological effects of vaccination and booster programmes together with an important assessment of socio-economic data regarding attitudes to uptake of vaccine.

The two important messages of the manuscript are that booster vaccinations are important and encouraging both adults and adolescents to get their doses is a very important public health message that is clearly effective in well educated adults and their adolescent children. 

My only concern about the study, though the authors do address this point, is the highly educated population that has been studied. Please could the authors give any figures as to the number of adults in the population as a whole who are educated to degree level and beyond? This is important as it is clear that the population examined in the study are compliant and aware of the severity of the COVID pandemic and the critical role of vaccination.  It is possible that increased public health messages may be more appropriately targeted at other non-degree educated population groups to increase the uptake of booster vaccinations.

Round 2

Reviewer 1 Report

This reviewer would have liked to see the two tables included in the author's response to the initial review included in the manuscript. Providing the data in these two tables in the manuscript would significantly improve the presentation of the results. However, I respect the author's choices not to make these types of revisions to the presentation of the results.